# Identified risk factors for dry eye syndrome: A systematic review and meta-analysis

Lijun Qian[1,2], Wei Wei[3]*

1 Department of Ophthalmology, Jinhua Hospital of Traditional Chinese Medicine, Jinhua, China, 2 Nanjing University of Chinese Medicine, Nanjing, China, 3 Department of Ophthalmology, Hospital Affiliated to Nanjing University of Chinese Medicine, Nanjing, China

* weix58@163.com

## Abstract

A meta-analytic approach was used to identify potential risk factors for dry eye syndrome. PubMed, Embase, and the Cochrane library were systematically searched for studies investigated the risk factors for dry eye syndrome from their inception until September 2021. The odds ratio (OR) with 95% confidence interval (CI) was calculated using the random-effects model. Forty-eight studies comprising 493,630 individuals were included. Older age (OR: 1.82; $P<0.001$), female sex (OR: 1.56; $P<0.001$), other race (OR: 1.27; $P<0.001$), visual display terminal use (OR: 1.32; $P<0.001$), cataract surgery (OR: 1.80; $P<0.001$), contact lens wear (OR: 1.74; $P<0.001$), pterygium (OR: 1.85; $P = 0.014$), glaucoma (OR: 1.77; $P = 0.007$), eye surgery (OR: 1.65; $P<0.001$), depression (OR: 1.83; $P<0.001$), post-traumatic stress disorder (OR: 1.65; $P<0.001$), sleep apnea (OR: 1.57; $P = 0.003$), asthma (OR: 1.43; $P<0.001$), allergy (OR: 1.38; $P<0.001$), hypertension (OR: 1.12; $P = 0.004$), diabetes mellitus (OR: 1.15; $P = 0.019$), cardiovascular disease (OR: 1.20; $P<0.001$), stroke (OR: 1.32; $P<0.001$), rosacea (OR: 1.99; $P = 0.001$), thyroid disease (OR: 1.60; $P<0.001$), gout (OR: 1.40; $P<0.001$), migraines (OR: 1.53; $P<0.001$), arthritis (OR: 1.76; $P<0.001$), osteoporosis (OR: 1.36; $P = 0.030$), tumor (OR: 1.46; $P<0.001$), eczema (OR: 1.30; $P<0.001$), and systemic disease (OR: 1.45; $P = 0.007$) were associated with an increased risk of dry eye syndrome. This study reported risk factors for dry eye syndrome, and identified patients at high risk for dry eye syndrome.

**Data Availability Statement:** All relevant data are within the paper and its Supporting Information files.

**Funding:** The authors received no specific funding for this work.

## Introduction

Dry eye syndrome (DES) is defined as a multifactorial disease of the tears and ocular surface that could cause discomfort and visual disturbance, with potential damage to the ocular surface. These symptoms could affect quality of life and activities of daily living [1, 2]. The prevalence of DES is increasing and is seen in nearly one in five adults. Thus, this needs more attention from ophthalmologists [3, 4]. The role of the tear film has already been demonstrated. It has been shown to provide lubrication to the eyes, as well as nutrition and oxygen, and eliminate debris from the ocular surface [5]. Moreover, individuals with dry eyes also suffer from systemic diseases [4]. However, the prevalence of dry eyes is often underestimated

**Competing interests:** The authors have declared that no competing interests exist.

**Abbreviations:** CI, confidence interval; COPD, chronic obstructive pulmonary disease; CVD, cardiovascular disease; DES, dry eye syndrome; DM, diabetes mellitus; HR, hazard ratio; MGD, meibomian gland dysfunction; NOS, Newcastle-Ottawa Scale; OR, odds ratio; PTSD, post-traumatic stress disorder; RR, relative risk; VDT, visual display terminal.

because of varying presentation and symptoms [6]. Studies have demonstrated that age and sex are significantly associated with increased risk of DES; however, the pathogenesis of DES is not fully understood [7, 8].

Several studies have already identified risk factors for DES. Major risk factors include older age, female sex, having undergone postmenopausal estrogen therapy or ocular surface surgery, and using antihistamine medications [9]. Moreover, the occupational risk factor of visual display terminal (VDT) use was related to the progression of DES, which could be explained by a decreased blink rate and increased proportion of incomplete blinks that could be caused by the increased exposure of the ocular surface to the environment. Outdoor environments, sunlight, and air pollution in tropical countries are also associated with an elevated risk of DES [10, 11]. Furthermore, other risk factors for DES include vitamin D deficiency and diabetes mellitus (DM) [12, 13]. However, whether the comorbidities of individuals could affect the risk of DES remained controversial. We, therefore, performed a systematic review and meta-analysis to independently identify risk factors for DES.

## Methods

### Data sources, search strategy, and selection criteria

The current study was performed and reported following the Preferred Reporting Items for Systematic Reviews and Meta-Analysis Statement [14]. Studies reporting the risk factors of DES were eligible in our study, and publication language was restricted to English. PubMed, Embase, and the Cochrane library were systematically searched for eligible studies from their inception until September 2021, and using the following text word or Medical Subject Heading terms: "dry eye syndrome", "dry eye disease", "Keratoconjunctivitis Sicca", "Xerophthalmia", and "Risk Factors". The details of search strategy in PubMed are listed in S1 File. The reference lists of relevant original and review articles were manually screened to identify further eligible studies.

Two reviewers (QL and WW) independently performed study assessment following a standardized approach. Any disagreement between reviewers was settled by discussion until a consensus was reached. A study was included if the following criteria were met: (1) it was a cross-sectional, retrospective, or prospective observational study; (2) risk factors were reported for ≥ 3 studies [15] and included such factors as age, sex, race, residence, education level, obesity, dyslipidemia, alcohol, smoking, VDT use, cataract surgery, contact lens wear, pterygium, glaucoma, age-related maculopathy, eye surgery, depression, post-traumatic stress disorder (PTSD), sleep apnea, asthma, allergy, hypertension, DM, cardiovascular disease (CVD), stroke, rosacea, thyroid disease, chronic obstructive pulmonary disease (COPD), gout, migraines, arthritis, osteoporosis, tumor, meibomian gland dysfunction (MGD), eczema, and systemic disease; and (3) it reported effect estimates (relative risk [RR], hazard ratio [HR], or odds ratio [OR]) and 95% confidence interval (CI) for risk factors of DES. Interventional study, animal study, review, and letter to editor was excluded.

### Data collection and quality assessment

Two reviewers (QL and WW) independently abstracted the following items, including study group or first author's name, publication year, country, study design, sample size, age, % of males, population status, % of DES cases, definition of DES, risk factors, adjusted factors, and reported effect estimates. The effect estimate with maximal adjustment for potential confounders was selected if a study reported several multivariable-adjusted effect estimates. Study quality was assessed using the Newcastle-Ottawa Scale (NOS), which has already been validated for assessing the quality of observational studies in meta-analysis [16]. A total of 8 items in 3

subscales were included in NOS. The star system in each study ranged from 0–9. Inconsistent results for the data abstracted and quality assessment between the two reviewers were settled following mutually discussion referred to the original article.

## Statistical analysis

Identified risk factors for DES were analyzed based on the OR, RR, or HR, with its 95% CI, in individual studies. Then the pooled ORs with 95%CI were calculated using the random-effects model [17, 18]. $I^2$ and Q statistic were applied to assess heterogeneity across included studies. Significant heterogeneity was defined as $I^2 > 50.0\%$ or $P < 0.10$ [19, 20]. Sensitivity analysis was performed for factors reported in $\geq 4$ studies to assess the robustness of pooled conclusion through sequentially removing individual studies [21]. Subgroup analyses were performed for factors reported in $\geq 4$ studies on the basis of the country. The difference between subgroups was assessed using the interaction $P$ test [22]. Visual inspections of funnel plots for factors reported in $\geq 4$ studies were performed to qualitatively assess publication bias. The Egger or Begg tests were used to quantitatively assess publication bias [23, 24]. The $P$-value for all pooled results was 2-sided, and the inspection level was 0.05. All of the statistical analysis in our study was performed using software STATA (version 12.0; Stata Corporation, College Station, TX, USA).

## Results

### Literature search

A total of 1,672 studies were identified from initial electronic searches. Details of the study selection process are presented in Fig 1. Of these, 912 articles were removed because they were duplicates. A further 671 articles were excluded owing to irrelevant titles or abstracts. The remaining 89 studies were retrieved for full-text evaluations, with 41 studies removed because

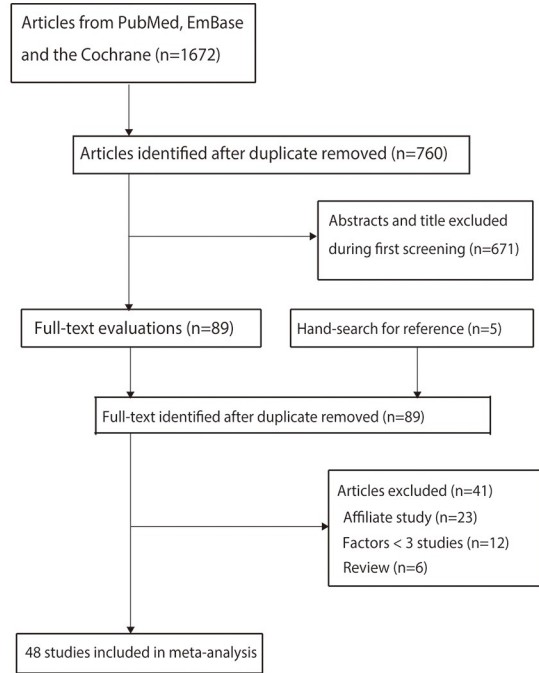

**Fig 1. Details of the literature search and study selection processes.**

of: affiliate study (n = 23), evaluated factors < 3 studies (n = 12), and review-type articles (n = 6). A manual search of the reference lists of relevant articles did not yield any additional studies. Finally, 48 studies were selected for the final meta-analysis [25–71]. Characteristics of the included studies and involved individuals are summarized in Table 1.

## Study characteristics

Of 48 included studies, 39 studies were designed as cross-sectional, 7 studies were designed as retrospective, and 2 studies designed as prospective. A total of 493,630 individuals were included, and the sample size ranged from 86 to 102,582. The mean age of included individuals ranged from 10.9 to 82.2. Twenty-nine studies were performed in Eastern countries, with the remaining 19 studies conducted in Western countries. Thirty-nine studies were population based. The remaining 9 studies were hospital based. The DES definition based on question-naire were reported in 33 studies, 10 studies used TBUT, ST, or FSS defined DES, 3 studies applied ICD9 code and the remaining 2 studies used clinician-diagnosed defined DES. Study quality was assessed using the NOS; 11 studies had 8 stars, 18 had 7 stars, and the remaining 19 had 6 stars (S1 Table). The quality of included studies mainly affect by the representativeness of the exposed cohort, and comparability on the basis of the design or analysis.

## Meta-analysis

**Demographic factors.** The number of studies that reported on the association of age, sex, and race as risk factors for DES was 15, 29, and 5, respectively (Fig 2 and S2 File). We noted that older adults (OR: 1.82; 95%CI: 1.47–2.26; $P<0.001$), females (OR: 1.56; 95%CI: 1.36–1.78; $P<0.001$), and those of other race (OR: 1.27; 95%CI: 1.11–1.44; $P<0.001$) had an increased risk of DES. There was significant heterogeneity for age ($I^2$ = 96.0%; $P<0.001$), sex ($I^2$ = 95.0%; $P<0.001$), and race ($I^2$ = 52.1%; $P = 0.080$). Sensitivity analysis indicated these pooled conclu-sions were robust and not altered by sequentially excluding individual studies (S3 File). The results of subgroup analyses were consistent with overall analysis when stratified according to the region (Table 2). There were no significant publication biases for age ($P$-value for Egger: 0.175; $P$-value for Begg: 1.000), sex ($P$-value for Egger: 0.417; $P$-value for Begg: 0.253), and race ($P$-value for Egger: 0.174; $P$-value for Begg: 0.806) regarding risk for DES (S4 File).

The number of studies reporting an association of residence, education level, obesity, and dyslipidemia regarding the risk of DES were 4, 8, 4, and 7, respectively (Fig 2 and S2 File). We noted that residence (urban versus rural) (OR: 1.41; 95%CI: 0.96–2.08; $P = 0.078$), education level (high versus low) (OR: 1.09; 95%CI: 0.88–1.34; $P = 0.443$), obesity (OR: 1.04; 95%CI: 0.87–1.24; $P = 0.671$), and dyslipidemia (OR: 1.18; 95%CI: 0.97–1.45; $P = 0.104$) were not asso-ciated with increased risk for DES. There was significant heterogeneity for residence ($I^2$ = 87.8%; $P<0.001$), education level ($I^2$ = 76.9%; $P<0.001$), and dyslipidemia ($I^2$ = 92.9%; $P<0.001$), while there was no evidence of heterogeneity for obesity ($I^2$ = 0.0%; $P = 0.530$). Sen-sitivity analyses indicated that residence, education level, and dyslipidemia might be associated with an elevated risk of DES, while the association between obesity and DES persisted (S3 File). Subgroup analyses demonstrated that education level and dyslipidemia were associated with an increased risk of DES when pooling studies conducted in Eastern countries (Table 2). No significant publication bias for residence ($P$-value for Egger: 0.875; $P$-value for Begg: 0.734), education level ($P$-value for Egger: 0.985; $P$-value for Begg: 0.902), and obesity ($P$-value for Egger: 0.638; $P$-value for Begg: 0.308) with the risk of DES was noted, whereas potential sig-nificant publication bias for dyslipidemia ($P$-value for Egger: 0.037; $P$-value for Begg: 1.000) with the risk of DES was seen (S4 File).

**Table 1. The baseline characteristics of included studies.**

| Study | Country | Study design | Sample size | Age (years) | Male (%) | Population | DES (%) | Definition of DES | Reported factors | Adjusted factors |
|---|---|---|---|---|---|---|---|---|---|---|
| BDES 2000 [25] | USA | C | 3,722 | 65.0 | 43.0 | PB | 14.4 | Questionnaire | DM, arthritis, TD, osteoporosis, gout, ES, CLW, alcohol, smoking | Age and sex |
| Lee 2002 [26] | Indonesia | C | 1,058 | 37.0 | 47.7 | PB | 27.5 | Questionnaire | Sex, smoking, pterygium | Sex, age, occupation, smoking, and pterygium |
| BMES 2003 [27] | Australia | C | 1,174 | 60.8 | 44.2 | PB | 57.5 | Questionnaire | Arthritis, asthma, DM, gout, smoking, alcohol | Age and sex |
| Sahai 2005 [28] | India | C | 500 | > 20.0 | 55.2 | HB | 18.4 | Questionnaire | Smoking | Age and sex |
| Nichols 2006 [29] | USA | C | 360 | 31.1 | 32.0 | HB | 55.3 | Questionnaire | Sex | Nominal water content, PLTF |
| Uchino 2008 [30] | Japan | C | 3,549 | 22.0–60.0 | 74.4 | PB | 10.1 | Questionnaire | Age, sex, VDT, systemic disease, smoking, contact lens | Age, gender, VDT use, systemic disease systemic medication, smoking, contact lens use |
| Lu 2008 [31] | China | C | 1,840 | 56.3 | 56.0 | PB | 52.4 | TFBT, ST, FSS | Age, education level, smoking alcohol | Crude |
| PHS 2009 [32] | USA | C | 25,444 | 64.4 | 100.0 | PB | 23.0 | Questionnaire | Age, race, hypertension, tumor, DM | Crude |
| TSES 2009 [33] | Spain | C | 654 | 63.6 | 37.2 | PB | 11.0 | Questionnaire | Sex, VDT use, CLW, rosacea, allergy, DM, hypertension, COPD, education level, alcohol, smoking | Age and sex |
| BES 2009 [34] | China | C | 1,957 | 56.5 | 43.1 | PB | 21.0 | Questionnaire | Sex, residence, glaucoma, MD, DM, hypertension, smoking, alcohol | Age, sex, region, undercorrection of refractive error, and nuclear cataract |
| THES 2010 [35] | China | C | 1,816 | 54.9 | 53.9 | PB | 50.1 | TBUT, ST, FSS | Pterygium, age, sex, education level, smoking, alcohol | Crude |
| Kim 2011 [36] | Korea | C | 650 | 71.9 | 48.3 | PB | 30.5 | Questionnaire | Sex, residence, depression, MGD | Crude |
| Koumi Study 2011 [37] | Japan | C | 2,791 | > 40.0 | 43.7 | PB | 16.5 | Questionnaire | Age, smoking, alcohol, BMI, education level, VDT use, CLW, stroke, CVD, hypertension, DM | Age, smoking, alcohol, BMI, education level, VDT use, CLW, stroke, CVD, hypertension, DM |
| USVAP 2011 [38] | USA | R | 16,862 | NA | NA | PB | 12.2 | ICD9 code | Sex, race, DM, hypertension, dyslipidemia, CVD, stroke, PTSD, depression, alcohol, arthritis, gout, TD, tumor, sleep apnea, rosacea, glaucoma | Age and sex |
| Zhang 2012 [39] | China | C | 1,885 | < 18.0 | 50.8 | PB | 23.7 | Questionnaire | CLW, sleep apnea | CLW, sleep apnea, myopia, inadequate refractive correction, topical ophthalmic medication |
| TNHRI 2012 [40] | China | R | 48,028 | 52.4 | 26.6 | PB | 25.0 | ICD9 code | Hypertension, CVD, dyslipidemia, stroke, migraines, arthritis, COPD, asthma, DM, TD, depression, and tumor | Age, sex, region, and incomes |
| TOS 2013 [41] | Japan | C | 561 | 43.3 | 66.7 | PB | 11.6 | Questionnaire | Sex, age, smoking, VDT use, CLW, systemic disease, hypertension | Sex, age, smoking, VDT use, CLW, systemic disease, hypertension |

*(Continued)*

**Table 1.** (Continued)

| Study | Country | Study design | Sample size | Age (years) | Male (%) | Population | DES (%) | Definition of DES | Reported factors | Adjusted factors |
|---|---|---|---|---|---|---|---|---|---|---|
| TwinUK 2014 [42] | UK | C | 3,824 | 57.1 | 0.0 | PB | 9.6 | Questionnaire | CLW, CS, glaucoma, MD, osteoporosis, asthma, allergy, TD, arthritis, dyslipidemia, hypertension, DM, cancer, stroke, migraine, depression | Age |
| KNHNES 2014 [43] | Korea | C | 11,666 | 49.9 | 42.8 | PB | 8.0 | Questionnaire | Age, sex, education level, residence, hypertension, obesity, dyslipidemia, arthritis, TD, smoking, alcohol, sleep apnea, ES | Age, sex, education level, residence, hypertension, obesity, dyslipidemia, arthritis, TD, smoking, alcohol, sleep apnea, ES |
| Moon 2014 [44] | Korea | C | 288 | 10.9 | 49.3 | PB | 9.7 | Questionnaire | VDT use | Age, and sex |
| BDOS 2014 [45] | USA | C | 3,275 | 49.0 | 45.4 | PB | 14.5 | Questionnaire | Age, sex, CLW, arthritis, allergies, TD, migraine | Age, and sex |
| TNHI 2015 [46] | China | R | 10,325 | 61.9 | 36.7 | PB | 20.0 | ICD9 code | DM, hypertension, dyslipidemia, CVD | DM, hypertension, dyslipidemia, CVD |
| Yang 2015 [47] | China | R | 1,908 | 56.2 | 41.4 | HB | 41.4 | TFBT, ST, and FSS | DM, arthritis, tumor, acne rosacea, PTSD, VDT use | DM, arthritis, tumor, acne rosacea, PTSD, VDT use |
| Tan 2015 [48] | Singapore | C | 1,004 | 38.2 | 44.1 | PB | 12.3 | Questionnaire | Sex, age, CLW, alcohol | Crude |
| Shah 2015 [49] | India | C | 400 | 58.6 | 48.0 | HB | 54.3 | TBUT | DM, ES, MGD | Occupation, indoor table work, DM previous ocular surgery, MGD |
| Olaniyan 2016 [50] | Nigeria | C | 363 | 59.1 | 48.2 | PB | 32.5 | Questionnaire | Age, ES | Age, work place, medication use, ocular surgery, postmenopausal state |
| Alshamrani 2017 [51] | Saudi Arabia | C | 1,858 | 39.3 | 48.0 | PB | 32.1 | Questionnaire | Sex, age, residence, smoking, CLW, DM, hypertension, asthma, CVD, TD, arthritis, gout, osteoporosis | Sex, age, residence, work status, smoking, currently wearing, and history of trachoma |
| NHWS 2017 [52] | USA | C | 73,211 | > 18.0 | 48.4 | PB | 6.9 | Questionnaire | Age, sex, race, education level | Age and sex |
| SMES 2017 [53] | Singapore | P | 1,682 | 56.9 | 44.6 | PB | 5.1 | Questionnaire | DM, hypertension, smoking, CLW, stroke, CVD, TD, glaucoma, MGD, pterygium | Sex, age, income, smoking, CLW, cataract surgery, thyroid disease |
| Gong 2017 [54] | China | C | 1,015 | 54.6 | 29.7 | PB | 27.8 | Questionnaire | VDT use, DM, hypertension, arthritis, smoking, alcohol | Sex, age, VDT use, DM, hypertension, arthritis, dry mouth, smoking, alcohol, and spicy diets |
| Asiedu 2017 [54] | Ghana | C | 650 | 22.0 | 66.6 | PB | 44.3 | Questionnaire | Age, sex, allergies, alcohol, VDT use | Age, sex, allergies, alcohol, VDT use |
| Graue-Hernandez 2018 [55] | Mexico | C | 1,508 | 64.7 | 40.3 | PB | 41.1 | Questionnaire | Sex, smoking, DM, alcohol, hypertension | Sex, smoking, DM, alcohol, hypertension |
| SES 2018 [56] | Spain | C | 264 | 56.8 | 32.7 | PB | 25.4 | TBUT, ST, FSS | Sex, education level, VDT use, alcohol, smoking, hypertension, DM, COPD, CVD, TD, rosacea | Age |
| Iglesias 2018 [57] | USA | R | 86 | 71.0 | 95.0 | HB | 32.1 | Questionnaire | Race, DM, depression, PTSD, sleep apnea, glaucoma | Crude |

(Continued)

**Table 1.** (Continued)

| Study | Country | Study design | Sample size | Age (years) | Male (%) | Population | DES (%) | Definition of DES | Reported factors | Adjusted factors |
|---|---|---|---|---|---|---|---|---|---|---|
| TMS 2018 [58] | France | C | 1,045 | 82.2 | 71.8 | PB | 34.4 | Questionnaire | Obesity, smoking, alcohol, education level, hypertension, DM, depression, CS, MD, glaucoma | Age, and sex |
| Shehadeh-Mashor 2019 [59] | Israel | R | 25,317 | 27.0 | 55.0 | PB | 6.0 | TBUT, and ST | Sex, CLW | Age and sex |
| Zhang 2019 [60] | China | C | 31,124 | NA | 49.1 | HB | 57.6 | ST, and FSS | Sex, age, DM, arthritis, TD, ES | Sex, age, refractive surgery |
| Yasir 2019 [61] | Saudi Arabia | C | 890 | > 40.0 | 55.5 | PB | 35.9 | Questionnaire | Glaucoma, DM, and hypertension | Crude |
| HTS 2019 [62] | Japan | C | 356 | 55.5 | 37.4 | PB | 33.4 | Questionnaire | Sex, smoking, CLW, hypertension | Sex, eye makeup use, smoking CLW, hypertension, sleeping pills |
| Hyon 2019 [63] | Korea | C | 232 | > 20.0 | 15.1 | PB | 42.7 | Questionnaire | Sex, VDT use | Sex, and VDT use |
| Ben-Eli 2019 [64] | Israel | R | 331 | 53.6 | 24.8 | HB | 36.3 | Clinician-diagnosed | Smoking, alcohol | Ethnicity, smoking, alcohol, hospitalization for infection |
| Yu 2019 [65] | China | C | 23,922 | NA | 48.8 | HB | 61.6 | TBUT, and FSS | Sex, age, ES, arthritis, TD | Humidity, air pressure, and air temperature |
| Rossi 2019 [66] | Italy | C | 194 | 41.8 | 34.5 | HB | 16.5 | TBUT, and FSS | Sex, VDT use | Age, sex, VDT use, visual acuity, and presbyopia |
| Wang 2020 [67] | New Zealand | C | 372 | 39.0 | 40.3 | PB | 29.0 | Clinician-diagnosed | Sex, CLW, anxiety, asthma, DM, depression, dyslipidemia, hypertension, cancer, migraine, TD, CS, ES | Age, CLW, ethnicity, migraine, menopause, systemic disease, thyroid disease, antidepressant medication, and oral contraceptive therapy |
| Shanti 2020 [68] | Palestine | C | 769 | 43.6 | 47.3 | PB | 64.0 | TBUT, ST, FSS | Sex, VDT use, smoking, DM, hypertension | Age, sex, VDT use, smoking, systemic disease |
| JPHC 2020 [69] | Japan | P | 102,582 | 58.3 | 46.2 | PB | 24.6 | Questionnaire | VDT use | Age, smoking, education status, income, and public health area |
| Alkabbani 2021 [70] | United Arab Emirates | C | 452 | > 17.0 | 36.3 | PB | 62.6 | Questionnaire | Age, sex, CLW, ES, VDT use, smoking | Age, sex, CLW, ES, VDT use, smoking |
| LCS 2021 [71] | Netherlands | C | 79,866 | 50.4 | 40.8 | PB | 9.1 | Questionnaire | Sex, CLW, MD, glaucoma, ES, CS, arthritis, gout, CVD, stroke, migraine, depression, PTSD, COPD, asthma, sleep apnea, rosacea, allergy, DM, osteoporosis, TD, anemia | Age, and sex |

*BMI: body mass index; C: cross-sectional; CLW: contact lens wear; COPD: chronic obstructive pulmonary disease; CS: cataract surgery; CVD: cardiovascular disease; DM: diabetes mellitus; ES: eye surgery; FSS: fluorescein staining score; HB: hospital-based; MD: macular degeneration; MDG: meibomian gland dysfunction; MI: myocardial infarction; NA: not available; P: prospective; PB: population-based; PLTF: prelens tear film; PTSD: post-traumatic stress disorder; R: retrospective; ST: Schirmer test; TBUT: tear film break-up time; TD: thyroid disease; TFBT: tear film breakup time; VDT: visual display terminal

The number of studies reporting an association of alcohol, smoking, and VDT use with the risk of DES was 15, 22, and 14, respectively (Fig 2 and S2 File). We noted that alcohol intake (OR: 0.98; 95%CI: 0.81–1.18; $P = 0.808$) and current smoking (OR: 1.00; 95%CI: 0.86–1.16; $P = 0.986$) were not associated with risk for DES, while VDT use was associated with an increased risk of DES (OR: 1.32; 95%CI: 1.17–1.49; $P<0.001$). There was significant

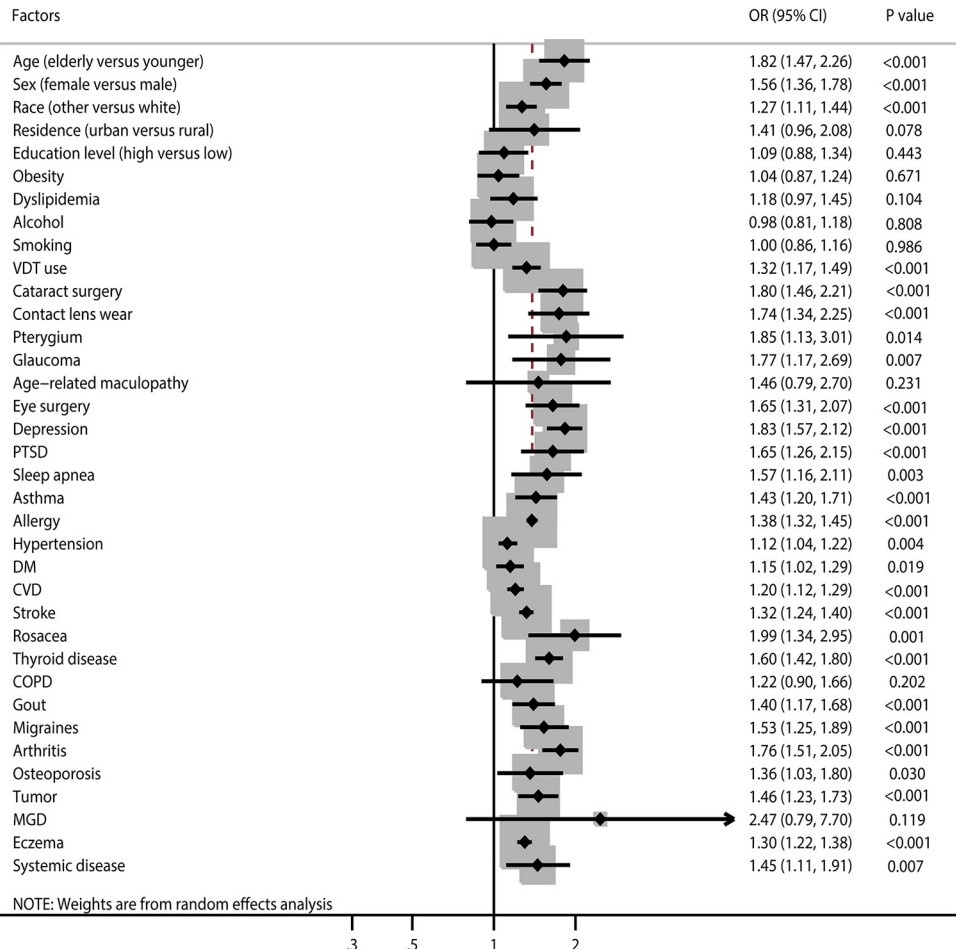

| Factors | OR (95% CI) | P value |
|---|---|---|
| Age (elderly versus younger) | 1.82 (1.47, 2.26) | <0.001 |
| Sex (female versus male) | 1.56 (1.36, 1.78) | <0.001 |
| Race (other versus white) | 1.27 (1.11, 1.44) | <0.001 |
| Residence (urban versus rural) | 1.41 (0.96, 2.08) | 0.078 |
| Education level (high versus low) | 1.09 (0.88, 1.34) | 0.443 |
| Obesity | 1.04 (0.87, 1.24) | 0.671 |
| Dyslipidemia | 1.18 (0.97, 1.45) | 0.104 |
| Alcohol | 0.98 (0.81, 1.18) | 0.808 |
| Smoking | 1.00 (0.86, 1.16) | 0.986 |
| VDT use | 1.32 (1.17, 1.49) | <0.001 |
| Cataract surgery | 1.80 (1.46, 2.21) | <0.001 |
| Contact lens wear | 1.74 (1.34, 2.25) | <0.001 |
| Pterygium | 1.85 (1.13, 3.01) | 0.014 |
| Glaucoma | 1.77 (1.17, 2.69) | 0.007 |
| Age–related maculopathy | 1.46 (0.79, 2.70) | 0.231 |
| Eye surgery | 1.65 (1.31, 2.07) | <0.001 |
| Depression | 1.83 (1.57, 2.12) | <0.001 |
| PTSD | 1.65 (1.26, 2.15) | <0.001 |
| Sleep apnea | 1.57 (1.16, 2.11) | 0.003 |
| Asthma | 1.43 (1.20, 1.71) | <0.001 |
| Allergy | 1.38 (1.32, 1.45) | <0.001 |
| Hypertension | 1.12 (1.04, 1.22) | 0.004 |
| DM | 1.15 (1.02, 1.29) | 0.019 |
| CVD | 1.20 (1.12, 1.29) | <0.001 |
| Stroke | 1.32 (1.24, 1.40) | <0.001 |
| Rosacea | 1.99 (1.34, 2.95) | 0.001 |
| Thyroid disease | 1.60 (1.42, 1.80) | <0.001 |
| COPD | 1.22 (0.90, 1.66) | 0.202 |
| Gout | 1.40 (1.17, 1.68) | <0.001 |
| Migraines | 1.53 (1.25, 1.89) | <0.001 |
| Arthritis | 1.76 (1.51, 2.05) | <0.001 |
| Osteoporosis | 1.36 (1.03, 1.80) | 0.030 |
| Tumor | 1.46 (1.23, 1.73) | <0.001 |
| MGD | 2.47 (0.79, 7.70) | 0.119 |
| Eczema | 1.30 (1.22, 1.38) | <0.001 |
| Systemic disease | 1.45 (1.11, 1.91) | 0.007 |

NOTE: Weights are from random effects analysis

**Fig 2. Summary results of risk factors for dry eye syndrome.**

heterogeneity for alcohol ($I^2$ = 62.2%; $P$ = 0.001), smoking ($I^2$ = 64.6%; $P$<0.001), and VDT use ($I^2$ = 80.1%; $P$<0.001). Sensitivity analysis indicated that alcohol intake might play an important role in the risk of DES, while the pooled results for the associations of smoking and VDT use with the risk of DES were robust (S3 File). The results of subgroup analyses were consistent with the overall analysis (Table 2). No significant publication bias for smoking ($P$-value for Egger: 0.569; $P$-value for Begg: 0.822) and VDT use ($P$-value for Egger: 0.370; $P$ value for Begg: 0.827) with the risk of DES was found, whereas potential significant publication bias for alcohol ($P$-value for Egger: 0.032; $P$-value for Begg: 0.921) with the risk of DES was noted (S4 File).

**Clinical characteristics.** The number of studies that reported on the association of cataract surgery, contact lens wear, pterygium, glaucoma, age-related maculopathy, and eye surgery with the risk of DES were 7, 17, 4, 9, 3, and 8, respectively (Fig 2 and S2 File). We noted that cataract surgery (OR: 1.80; 95%CI: 1.46–2.21; $P$<0.001), contact lens wear (OR: 1.74; 95% CI: 1.34–2.25; $P$<0.001), pterygium (OR: 1.85; 95%CI: 1.13–3.01; $P$ = 0.014), glaucoma (OR: 1.77; 95%CI: 1.17–2.69; $P$ = 0.007), and eye surgery (OR: 1.65; 95%CI: 1.31–2.07; $P$<0.001) were associated with an increased risk of DES, while age-related maculopathy was not associated with risk of DES (OR: 1.46; 95%CI: 0.79–2.70; $P$ = 0.231). Significant heterogeneity was noted for cataract surgery ($I^2$ = 64.8%; $P$<0.001), contact lens wear ($I^2$ = 93.5%; $P$<0.001),

**Table 2. Subgroup analyses according to region.**

| Factors | Subgroup | OR and 95%CI | *P* value | $I^2$ (%) | $P_{heterogeneity}$ | *P* value between subgroups |
|---|---|---|---|---|---|---|
| Age (elderly versus younger) | Eastern countries | 1.78 (1.46–2.19) | < 0.001 | 89.4 | < 0.001 | < 0.001 |
| | Western countries | 2.04 (1.05–3.97) | 0.036 | 98.7 | < 0.001 | |
| Sex (female versus male) | Eastern countries | 1.53 (1.36–1.72) | < 0.001 | 84.0 | < 0.001 | < 0.001 |
| | Western countries | 1.52 (1.20–1.92) | < 0.001 | 95.8 | < 0.001 | |
| Race (other versus white) | Eastern countries | - | - | - | - | - |
| | Western countries | 1.27 (1.11–1.44) | < 0.001 | 52.1 | 0.080 | |
| Residence (urban versus rural) | Eastern countries | 1.41 (0.96–2.08) | 0.078 | 87.8 | < 0.001 | - |
| | Western countries | - | - | - | - | |
| Education level (high versus low) | Eastern countries | 1.28 (1.01–1.63) | 0.041 | 60.2 | 0.057 | 0.007 |
| | Western countries | 0.86 (0.56–1.33) | 0.505 | 80.7 | 0.001 | |
| Obesity | Eastern countries | 1.02 (0.83–1.25) | 0.866 | 2.1 | 0.360 | 0.685 |
| | Western countries | 1.11 (0.77–1.60) | 0.576 | - | - | |
| Dyslipidemia | Eastern countries | 1.35 (1.01–1.80) | 0.046 | 94.7 | < 0.001 | < 0.001 |
| | Western countries | 1.05 (0.82–1.35) | 0.676 | 77.2 | 0.004 | |
| Alcohol | Eastern countries | 1.04 (0.90–1.20) | 0.589 | 0.0 | 0.429 | 0.177 |
| | Western countries | 0.92 (0.64–1.32) | 0.641 | 76.1 | <0.001 | |
| Smoking | Eastern countries | 0.96 (0.81–1.15) | 0.668 | 65.2 | < 0.001 | 0.046 |
| | Western countries | 1.09 (0.82–1.45) | 0.554 | 60.0 | 0.020 | |
| VDT use | Eastern countries | 1.33 (1.17–1.53) | < 0.001 | 85.5 | < 0.001 | 0.436 |
| | Western countries | 1.33 (1.06–1.68) | 0.015 | 0.0 | 0.457 | |
| Cataract surgery | Eastern countries | 2.16 (1.62–2.89) | < 0.001 | 0.0 | 0.792 | 0.561 |
| | Western countries | 1.69 (1.28–2.21) | < 0.001 | 76.0 | 0.002 | |
| Contact lens wear | Eastern countries | 2.01 (1.48–2.71) | <0.001 | 72.8 | <0.001 | 0.003 |
| | Western countries | 1.41 (0.93–2.14) | 0.105 | 97.1 | <0.001 | |
| Pterygium | Eastern countries | 1.85 (1.13–3.01) | 0.014 | 89.0 | < 0.001 | - |
| | Western countries | - | - | - | - | |
| Glaucoma | Eastern countries | 2.15 (1.29–3.58) | 0.003 | 26.1 | 0.255 | 0.516 |
| | Western countries | 1.57 (0.92–2.68) | 0.098 | 96.5 | < 0.001 | |
| Age-related maculopathy | Eastern countries | 0.31 (0.07–1.35) | 0.118 | - | - | 0.007 |
| | Western countries | 1.91 (1.21–3.01) | 0.005 | 62.9 | 0.067 | |
| Eye surgery | Eastern countries | 1.62 (1.23–2.14) | 0.001 | 93.6 | <0.001 | <0.001 |
| | Western countries | 1.82 (1.39–2.37) | < 0.001 | 32.1 | 0.229 | |
| Depression | Eastern countries | 2.12 (1.95–2.32) | < 0.001 | 0.0 | 0.876 | < 0.001 |
| | Western countries | 1.66 (1.43–1.93) | < 0.001 | 67.0 | 0.010 | |
| PTSD | Eastern countries | 1.45 (1.04–2.01) | 0.027 | - | - | 0.121 |
| | Western countries | 1.71 (1.19–2.46) | 0.004 | 53.0 | 0.119 | |
| Sleep apnea | Eastern countries | 1.22 (1.11–1.35) | < 0.001 | 4.5 | 0.370 | < 0.001 |
| | Western countries | 2.17 (1.95–2.41) | < 0.001 | 0.0 | 0.749 | |
| Asthma | Eastern countries | 1.19 (0.98–1.45) | 0.076 | 29.0 | 0.235 | < 0.001 |
| | Western countries | 1.62 (1.49–1.77) | < 0.001 | 0.0 | 0.869 | |
| Allergy | Eastern countries | - | - | - | - | - |
| | Western countries | 1.38 (1.32–1.45) | < 0.001 | 0.0 | 0.418 | |
| Hypertension | Eastern countries | 1.06 (0.95–1.17) | 0.306 | 63.7 | 0.001 | 0.005 |
| | Western countries | 1.27 (1.14–1.41) | < 0.001 | 24.8 | 0.231 | |
| DM | Eastern countries | 1.20 (1.06–1.37) | 0.005 | 79.0 | < 0.001 | < 0.001 |
| | Western countries | 1.08 (0.87–1.34) | 0.460 | 88.5 | < 0.001 | |

(*Continued*)

**Table 2.** (Continued)

| Factors | Subgroup | OR and 95%CI | P value | $I^2$ (%) | $P_{heterogeneity}$ | P value between subgroups |
|---|---|---|---|---|---|---|
| CVD | Eastern countries | 1.26 (1.15–1.39) | < 0.001 | 0.0 | 0.753 | 0.084 |
| | Western countries | 1.15 (1.00–1.32) | 0.049 | 18.5 | 0.293 | |
| Stroke | Eastern countries | 1.31 (1.22–1.41) | < 0.001 | 0.0 | 0.978 | 0.667 |
| | Western countries | 1.35 (1.20–1.51) | < 0.001 | 0.0 | 0.589 | |
| Rosacea | Eastern countries | 3.75 (1.97–7.12) | < 0.001 | - | - | 0.032 |
| | Western countries | 1.74 (1.20–2.52) | 0.004 | 53.1 | 0.094 | |
| Thyroid disease | Eastern countries | 1.57 (1.29–1.91) | < 0.001 | 86.0 | <0.001 | 0.752 |
| | Western countries | 1.64 (1.45–1.84) | < 0.001 | 26.9 | 0.223 | |
| COPD | Eastern countries | 1.06 (0.84–1.34) | 0.625 | - | - | 0.006 |
| | Western countries | 1.37 (1.00–1.89) | 0.051 | 23.2 | 0.272 | |
| Gout | Eastern countries | 1.56 (0.70–3.49) | 0.275 | 83.3 | 0.014 | 0.175 |
| | Western countries | 1.34 (1.17–1.53) | < 0.001 | 0.0 | 0.860 | |
| Migraines | Eastern countries | 1.76 (1.57–1.98) | < 0.001 | - | - | < 0.001 |
| | Western countries | 1.41 (1.19–1.68) | < 0.001 | 54.2 | 0.088 | |
| Arthritis | Eastern countries | 1.74 (1.31–2.29) | < 0.001 | 95.6 | < 0.001 | 0.776 |
| | Western countries | 1.80 (1.57–2.07) | < 0.001 | 74.7 | 0.001 | |
| Osteoporosis | Eastern countries | 0.81 (0.51–1.29) | 0.377 | - | - | 0.004 |
| | Western countries | 1.53 (1.21–1.93) | < 0.001 | 75.8 | 0.016 | |
| Tumor | Eastern countries | 2.27 (0.83–6.22) | 0.111 | 94.7 | < 0.001 | 0.339 |
| | Western countries | 1.33 (1.17–1.50) | <0.001 | 39.5 | 0.175 | |

pterygium ($I^2$ = 89.0%; $P<0.001$), glaucoma ($I^2$ = 93.4%; $P<0.001$), age-related maculopathy ($I^2$ = 76.5%; $P$ = 0.005), and eye surgery ($I^2$ = 94.0%; $P<0.001$) with the risk of DES. Sensitivity analyses indicated that the pooled results for the association of cataract surgery, contact lens wear, pterygium, glaucoma, and eye surgery with the risk of DES persisted, whereas age-related maculopathy might be associated with the risk of DES (S3 File). Although most results in the subgroup analyses were consistent with the overall analysis, we noted that contact lens wear and glaucoma were not associated with the risk of DES when pooling studies performed in Western countries. Moreover, age-related maculopathy was associated with an increased risk of DES when pooling studies conducted in Western countries (Table 2). There was no significant publication bias for the association of cataract surgery (*P*-value for Egger: 0.194; *P*-value for Begg: 0.548), contact lens wear (*P*-value for Egger: 0.791; *P*-value for Begg: 0.387), pterygium (*P*-value for Egger: 0.681; *P*-value for Begg: 0.734), glaucoma (*P*-value for Egger: 0.950; *P*-value for Begg: 0.917), and eye surgery (*P*-value for Egger: 0.760; *P*-value for Begg: 0.266) with the risk of DES, while potential significant publication bias was noted for age-related maculopathy (*P*-value for Egger: 0.017; *P*-value for Begg: 0.308) with the risk of DES (S4 File).

**Comorbidities.** The number of studies that reported on the association of depression, PTSD, sleep apnea, asthma, and allergy with the risk of DES were 9, 4, 7, 6, and 6, respectively (Fig 2 and S2 File). We noted that depression (OR: 1.83; 95%CI: 1.57–2.12; $P<0.001$), PTSD (OR: 1.65; 95%CI: 1.26–2.15; $P<0.001$), sleep apnea (OR: 1.57; 95%CI: 1.16–2.11; $P$ = 0.003), asthma (OR: 1.43; 95%CI: 1.20–1.71; $P<0.001$), and allergy (OR: 1.38; 95%CI: 1.32–1.45; $P<0.001$) were associated with an increased risk of DES. There was significant heterogeneity for depression ($I^2$ = 80.7%; $P<0.001$), PTSD ($I^2$ = 55.0%; $P$ = 0.083), sleep apnea ($I^2$ = 91.5%; $P<0.001$), and asthma ($I^2$ = 76.5%; $P$ = 0.001), while no evidence of heterogeneity for allergy was observed ($I^2$ = 0.0%; $P$ = 0.418). Sensitivity analyses indicated that pooled conclusions for

the association of depression, PTSD, sleep apnea, asthma, and allergy with the risk of DES were stable after sequentially removing individual studies (S3 File). The results of subgroup analyses were consistent with overall analysis, except that asthma was not associated with the risk of DES if pooled studies were performed in Eastern countries (Table 2). No significant publication bias for the role of depression (*P*-value for Egger: 0.679; *P*-value for Begg: 0.348), PTSD (*P*-value for Egger: 0.415; *P*-value for Begg: 0.734), sleep apnea (*P*-value for Egger: 0.959; *P*-value for Begg: 0.764), asthma (*P*-value for Egger: 0.949; *P*-value for Begg: 1.000), and allergy (*P*-value for Egger: 0.189; *P*-value for Begg: 0.707) with DES were observed (S4 File).

The number of studies reporting on the association of hypertension, DM, CVD, stroke, rosacea, thyroid disease, and COPD with the risk of DES were 21, 24, 8, 7, 5, 14, and 4, respectively (Fig 2 and S2 File). We noted that hypertension (OR: 1.12; 95%CI: 1.04–1.22; $P = 0.004$), DM (OR: 1.15; 95%CI: 1.02–1.29; $P = 0.019$), CVD (OR: 1.20; 95%CI: 1.12–1.29; $P<0.001$), stroke (OR: 1.32; 95%CI: 1.24–1.40; $P<0.001$), rosacea (OR: 1.99; 95%CI: 1.34–2.95; $P = 0.001$), and thyroid disease (OR: 1.60; 95%CI: 1.42–1.80; $P<0.001$) were associated with an increased risk of DES, while COPD was not associated with risk of DES (OR: 1.22; 95%CI: 10.90–1.66; $P = 0.202$). There was significant heterogeneity for hypertension ($I^2 = 60.2\%$; $P<0.001$), DM ($I^2 = 86.7\%$; $P<0.001$), rosacea ($I^2 = 63.6\%$; $P = 0.027$), thyroid disease ($I^2 = 74.6\%$; $P<0.001$), and COPD ($I^2 = 70.6\%$; $P = 0.017$), while no significant heterogeneity was observed for CVD ($I^2 = 4.8\%$; $P = 0.393$) and stroke ($I^2 = 0.0\%$; $P = 0.964$). The pooled conclusions for the association of hypertension, CVD, stroke, rosacea, and thyroid disease with the risk of DES were stable, while the conclusions for DM and COPD with DES were variable (S3 File). Although the results of subgroup analyses were consistent with the overall analysis in most subsets, we noted that hypertension was not related to DES if pooling in Eastern country studies, while DM was not associated with the risk of DES if pooled studies were performed in Western countries (Table 2). There was no significant publication bias for hypertension (*P*-value for Egger: 0.331; *P*-value for Begg: 0.928), DM (*P*-value for Egger: 0.765; *P*-value for Begg: 0.862), CVD (*P*-value for Egger: 0.357; *P*-value for Begg: 0.711), stroke (*P*-value for Egger: 0.485; *P*-value for Begg: 0.368), rosacea (*P*-value for Egger: 0.759; *P*-value for Begg: 0.806), thyroid disease (*P*-value for Egger: 0.996; *P*-value for Begg: 0.228), and COPD (*P*-value for Egger: 0.267; *P*-value for Begg: 1.000) (S4 File).

The number of studies reporting on the association of gout, migraines, arthritis, osteoporosis, tumor, MGD, eczema, and systemic disease with the risk of DES was 6, 5, 13, 4, 6, 3, 3, and 3, respectively (Fig 2 and S2 File). We noted that gout (OR: 1.40; 95%CI: 1.17–1.68; $P<0.001$), migraines (OR: 1.53; 95%CI: 1.25–1.89; $P<0.001$), arthritis (OR: 1.76; 95%CI: 1.51–2.05; $P<0.001$), osteoporosis (OR: 1.36; 95%CI: 1.03–1.80; $P = 0.030$), tumor (OR: 1.46; 95%CI: 1.23–1.76; $P<0.001$), eczema (OR: 1.30; 95%CI: 1.22–1.38; $P<0.001$), and systemic disease (OR: 1.45; 95%CI: 1.11–1.91; $P = 0.007$) were associated with an increased risk of DES, while MGD was not associated with risk of DES (OR: 2.47; 95%CI: 0.79–7.70; $P = 0.119$). There was significant heterogeneity for migraines ($I^2 = 86.4\%$; $P<0.001$), arthritis ($I^2 = 92.4\%$; $P<0.001$), osteoporosis ($I^2 = 82.1\%$; $P = 0.001$), tumor ($I^2 = 79.9\%$; $P<0.001$), and MGD ($I^2 = 85.2\%$; $P = 0.001$), while no significant heterogeneity for gout ($I^2 = 41.8\%$; $P = 0.126$), eczema ($I^2 = 0.0\%$; $P = 0.609$), and systemic disease ($I^2 = 0.0\%$; $P = 0.007$) was observed. The pooled conclusions for the association of gout, migraines, arthritis, osteoporosis, and tumor with the risk of DES were robust after sequentially removing individual studies (S3 File). Although the results of subgroup analyses were consistent with the overall analysis in most subsets, gout, osteoporosis, and tumor were not associated with risk of DES if pooled studies were performed in Eastern countries. There was no significant publication bias for gout (*P*-value for Egger: 0.902; *P*-value for Begg: 0.707), migraines (*P*-value for Egger: 0.249; *P*-value for Begg: 0.806), arthritis

(*P*-value for Egger: 0.169; *P*-value for Begg: 0.360), osteoporosis (*P*-value for Egger: 0.137; *P*-value for Begg: 0.308), and tumor (*P*-value for Egger: 0.721; *P*-value for Begg: 1.000) (S4 File).

## Discussion

This systematic review and meta-analysis was based on published observational studies explored potential risk factors for DES and included 493,630 individuals from 48 studies. We found that risk factors for DES included older age, female sex, other race, VDT use, cataract surgery, contact lens wear, pterygium, glaucoma, eye surgery, depression, PTSD, sleep apnea, asthma, allergy, hypertension, DM, CVD, stroke, rosacea, thyroid disease, gout, migraines, arthritis, osteoporosis, tumor, eczema, and systemic disease. Moreover, country of origin could affect association for age, sex, education level, dyslipidemia, smoking, contact lens wear, age-related maculopathy, eye surgery, depression, sleep apnea, asthma, hypertension, DM, rosacea, COPD, migraines, and osteoporosis regarding the risk of DES.

This current study primarily identified potential risk factors for DES, although several factors have already been demonstrated in individual studies. Prior studies have demonstrated that a 5-year incidence of dry eye rises from 10.7% to 17.9% alongside increasing age [72]. A potential reason could be the reduction of tear secretion with biological aging [2, 73]. Moreover, the sex difference in DES could be explained by various hormonal effects on the ocular surface and lacrimal gland [8]. The potential impact for VDT use could be due to increasing rates of incomplete blinks and accelerated evaporation of the tear film [74]. The increased risk of DES after cataract surgery could be explained by cataract surgery inducing tear film dysfunction [75]. The role of contact lens wear on DES could be explained in that placing a lens on the eye could cause disturbance of the tear film [76]. DES could be considered as a precipitating factor of primary pterygium [77]. The treatment of glaucoma could alter the surface of the eye through disturbing tear secretion, which could affect the progression of DES [78]. Studies have already found that open eye surgery could affect altered tear secretion in nearly 91% of patients, thus playing an important role in the risk of DES [79]. The potential role of depression and PTSD could be explained by the dysregulation of neuropeptides coupled with serotonin in human tears and serotonin receptors in human conjunctivae [80]. Sleep apnea is significantly associated with neuropathic pain, which could induce the progression of dry eye syndrome [81]. The role of asthma and allergy on the risk of DES could be explained by anti-histaminic and anti-inflammatory agents used for asthma and allergy treatment, which could potentially cause an elevated risk of DES [82].

This study found that hypertension and DM were associated with an increased risk of DES, which was consistent with the results of a prior meta-analysis [83]. A potential reason for this could be hypertension was not direct affect the risk of DES, while the use of anti-hypertensive medication could increase the risk of DES [33]. In addition, the risk of DES were not increased in hypertensive patients treated with anti-hypertensive medications, such as Angiotension Converting Enzyme inhibitors might play a protective role on the risk of DES [34]. Moreover, DM could induce a decrease in corneal sensation and tear production, impaired metabolic activity, and loss of cytoskeletal structure, all of which could affect the progression of DES [84]. The underlying therapies for CVD, stroke, and tumor could be regarded as disposing of factors for DES [25]. Rosacea is a well known risk factor for DES due to is pro-inflammatory effects that induce meibomian gland dysfunction and evaporative DES [85]. Studies have already found that thyroid disease is significantly related to ocular surface damage, eyelid retraction/impaired Bell's phenomenon, and reduced tear production [86]. Gout was associated with the tophaceous deposits in different locations of the eye, including eyelids, conjunctiva, cornea, iris, sclera, and orbit, a similar reason could explain the role of arthritis on DES [87]. The role

of migraines on DES could be explained by an inflammatory status in migraine patients potentially activating inflammation in the eyes [88]. The inflammation and hormone imbalance caused by osteoporosis could explain an elevated risk of DES [89]. The treatment for eczema and systemic disease could cause an elevated risk of DES [90].

Our study found that potential associations for age, sex, education level, dyslipidemia, smoking, contact lens wear, age-related maculopathy, eye surgery, depression, sleep apnea, asthma, hypertension, DM, rosacea, COPD, migraines, and osteoporosis with the risk of DES could be affected by country of origin. The disease distribution for DES is different in Eastern and Western countries, and the health policy in various countries could further affect the progression of DES. Moreover, environmental, dietary, and lifestyle factors among various countries differ, which could affect the progression of DES [91, 92].

Several shortcomings of this study should be acknowledged. First, this study contained cross-sectional, retrospective, and prospective observational studies, and the causality relationships between risk factors and DES could not available. Second, the heterogeneity for most risk factors was substantial, which was not fully explained by sensitivity and subgroup analyses. Third, the comorbidity and underlying therapies for individuals were not fully adjusted, which could affect the progression of DES. Fourth, the cutoff value for age, and definition for systemic disease, eye surgery, and DES are different across included studies, which could induce potential uncontrolled biases. Fifth, the climate type could affect the progression of DES, and nearly all of included studies did not address the climate type. Sixth, the analysis based on published articles, the gray literature and unpublished data were not available, and the publication bias was inevitable. Seventh, the analysis using the pooled data, and the detailed analyses were restricted. Finally, this study was not registered in PROSPERO, and the transparency was restricted.

## Conclusions

This study identified comprehensive risk factors for DES, including older age, female sex, other race, VDT use, cataract surgery, contact lens wear, pterygium, glaucoma, eye surgery, depression, PTSD, sleep apnea, asthma, allergy, hypertension, DM, CVD, stroke, rosacea, thyroid disease, gout, migraines, arthritis, osteoporosis, tumor, eczema, and systemic disease. Further large-scale prospective cohort studies should be performed to verify the results of this study.

## Supporting information

**S1 Checklist. PRISMA 2020 checklist.**
(PDF)

**S1 Table. Quality scores of prospective cohort studies using Newcastle-Ottawa Scale.**
(DOCX)

**S1 File. Search strategy in PubMed.**
(DOCX)

**S2 File. Forest plots for the risk factors of dry eye syndrome.**
(DOCX)

**S3 File. Sensitivity analyses for the risk factors of dry eye syndrome.**
(DOCX)

**S4 File. Funnel plots for the risk factors of dry eye syndrome.**
(DOCX)

## Author Contributions

**Conceptualization:** Lijun Qian, Wei Wei.

**Data curation:** Lijun Qian.

**Formal analysis:** Lijun Qian.

**Investigation:** Lijun Qian.

**Project administration:** Wei Wei.

**Resources:** Wei Wei.

**Software:** Lijun Qian.

**Supervision:** Lijun Qian.

**Validation:** Lijun Qian.

**Visualization:** Lijun Qian.

**Writing – original draft:** Lijun Qian, Wei Wei.

**Writing – review & editing:** Lijun Qian, Wei Wei.

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
