## [Decision Letter · Decision Letter 0]

19 Jan 2022

PONE-D-21-35192Identified risk factors for dry eye syndrome: A systematic review and meta-analysisPLOS ONE

Dear Dr. Wei,

Thank you for submitting your manuscript to PLOS ONE. After careful consideration, we feel that it has merit but does not fully meet PLOS ONE’s publication criteria as it currently stands. Therefore, we invite you to submit a revised version of the manuscript that addresses the points raised during the review process.

We look forward to receiving your revised manuscript.

Kind regards,

Michael Mimouni

Academic Editor

PLOS ONE

Journal Requirements:

2. Please confirm that you have included all items recommended in the PRISMA checklist including:

- the full electronic search strategy used to identify studies with all search terms and limits for at least one database.

- a Supplemental file of the results of the individual components of the quality assessment, not just the overall score, for each study included.

4. PLOS requires an ORCID iD for the corresponding author in Editorial Manager on papers submitted after December 6th, 2016. Please ensure that you have an ORCID iD and that it is validated in Editorial Manager. To do this, go to ‘Update my Information’ (in the upper left-hand corner of the main menu), and click on the Fetch/Validate link next to the ORCID field. This will take you to the ORCID site and allow you to create a new iD or authenticate a pre-existing iD in Editorial Manager. Please see the following video for instructions on linking an ORCID iD to your Editorial Manager account: https://www.youtube.com/watch?v=_xcclfuvtxQ.

Reviewers' comments:

Reviewer's Responses to Questions

**Comments to the Author**

1. Is the manuscript technically sound, and do the data support the conclusions?

Reviewer #1: Partly

Reviewer #2: Yes

2. Has the statistical analysis been performed appropriately and rigorously? 

Reviewer #1: Yes

Reviewer #2: Yes

3. Have the authors made all data underlying the findings in their manuscript fully available?

Reviewer #1: Yes

Reviewer #2: Yes

4. Is the manuscript presented in an intelligible fashion and written in standard English?

Reviewer #1: Yes

Reviewer #2: Yes

5. Review Comments to the Author

Reviewer #1: Manuscript ID: PONE-D-21-35192

Title: Identified risk factors for dry eye syndrome: A systematic review and meta-analysis

The authors report on a meta-analysis of an important topic – risk factors for dry eye syndrome. I thank the authors for their persistence with this large and complex analysis, however their analysis is limited by multiple methodologic and reporting concerns. The quality of evidence used to make these conclusions is also suspect, given the large proportion of cross-sectional studies that are susceptible to recall bias and variability in the adjustment for confounding factors across studies.

Specific comments are provided below:

1. Methods, Abstract: The model used for meta-analysis should be specified, as well as the inclusion and exclusion criteria for study selection.

2. Conclusion, Abstract: The authors should elaborate on their conclusion section to discuss the implications of their findings and/or areas for future research.

3. Introduction: expand on the risk factor ‘visual display terminal’ for readers who may be unfamiliar with this term.

4. Introduction: “However, the presence of other comorbidities and individuals’ characteristics on the risk of DES were not illustrated.” The authors should specify that there is conflicting or negative evidence for these other characteristics as opposed to these being uninvestigated characteristics.

5. Methods: “The following search terms were used: ("Dry Eye Syndromes"[Mesh]) AND "Risk Factors"[Mesh].” The search strategy provided by the authors is quite simplistic and does not account for other formulations of the same concept found in these major databases. No text-based searches are also integrated to ensure maximal literature coverage.

6. Methods: Did the authors investigate the gray literature as part of this systematic review?

7. Methods: specify with initials the authors that independently screened studies and extracted data.

8. Methods: the authors note that all studies reporting on dry eye risk factors were considered. Were interventional studies assessing the impact of dry eye treatment also included?

9. Methods: was the systematic review prospectively registered in the PROSPERO database?

10. Methods: The authors note that studies which assessed risk factors in at least 3 cohorts were included. Some studies assessing risk factors for dry eye may have only focused on one risk factor and would therefore be excluded under the current methodology. This overly restrictive inclusion criterion seems arbitrary and unwarranted.

11. Methods: Was a GRADE evaluation performed to assess the certainty of findings?

12. Methods: “Identified risk factors for DES were analyzed based on the effect estimate, with its 95% CI, in 99 individual studies.” What effect estimate was considered? Elaborate on the exact methods used. For continuous variables like age, how were they categorized into binary groups for analysis?

13. Methods: Only one subgroup analysis based on country was considered in the analysis. There are likely multiple potential confounders in this analysis that can be investigated for using subgroup analysis, including severity of dry eye, individuals with dry eye secondary to another condition eg Sjogren’s, evaporative vs aqueous deficient dry eye, study design, etc.

14. Methods, statistical analysis: specify the measure of association used to report the results – ie odds ratio with 95% confidence interval.

15. Results: what proportion of results of studies was derived from a multivariable analysis that controlled for confounding parameters versus a univariable analysis?

16. Results: provide more information in the study characteristics section on the type and severity of dry eye, observational vs interventional studies, the recency of publication, the pooled gender and age distribution, and other relevant baseline parameters for this analysis.

17. Results: expand on the results of the risk of bias assessment – what was the most frequent reason for downgrading the risk of bias?

18. Results: It would be important to clarify the definition of DES used in the analysis, and whether this definition was different across included studies.

19. Results: When reporting the results for each risk factor, it would be helpful to provide the proportion of the characteristic in the DES vs no DES groups, e.g. mean age, % female, race distribution, etc.

20. Results: For endpoints that had significant statistical heterogeneity, was the heterogeneity changed in the subgroup analysis?

21. Results: For glaucoma as a risk factor, it would important to clarify the therapy received, as this likely impacts the risk of DES.

22. Results: For eye surgery as a risk factor, clarify what type of surgeries were considered.

23. Results: In the reporting of the various risk factors, certain risk factors are grouped with others in the same paragraph. It is unclear how the various risk factors were grouped. It would be helpful to add subheadings to clarify this.

24. Results: “systemic disease” as a risk factor is not helpful or clinically relevant. This needs to be specified as to which specific systemic disease was considered or otherwise removed from the analysis.

25. Discussion: “we noted that other races versus white race were associated with an increased risk of DES, which is significantly related to the climate type.” Is race a risk factor independent of climate type? Elaborate on how climate type mediated this effect.

26. Discussion: “anti-hypertensive medication could increase the risk of DES”. What is the mechanism of this association?

27. Discussion: “The role of rosacea could be explained by its significant relation to corneal neovascularization and perforation, which could induce vision loss and ocular comorbidities.[85]” This relation is poorly described. Rosacea is a well known risk factor for DES due to is pro-inflammatory effects that induce meibomian gland dysfunction and evaporative DES.

28. Discussion: “Studies have already found that thyroid disease is significantly related to ocular surface damage, elevated lip aperture” – ‘elevated lip aperture’ should read ‘eyelid retraction’

29. Discussion: a major limitation is that the impact of confounding variables is unaddressed by certain studies, and multivariable associations were only reported for certain risk factors. Given the large proportion of cross-sectional studies, recall bias is likely a significant issue in these results.

30. Figure 2: p-value column – unsure why the p-values displayed in this way, as if multiple numbers are written on top of one another? What is the significance of the blue/gray background behind the forest plot for each endpoint?

Reviewer #2: This is a very well written meta-analysis regarding the potential risk factors for dry eye syndrome. Nevertheless, risk factors for DES are widely known, therefore, my question to the authors is:

In which way this meta analysis contributes to the literature already published regarding the risk factors for DES.

6. PLOS authors have the option to publish the peer review history of their article (what does this mean?). If published, this will include your full peer review and any attached files.

Reviewer #1: No

Reviewer #2: No

---

## [Author Response · Author response to Decision Letter 0]

21 Feb 2022

Point-By-Point Response

Journal Requirements:

Question 1: Please ensure that your manuscript meets PLOS ONE's style requirements, including those for file naming. The PLOS ONE style templates can be found at

Response: Thanks for this suggestion, and the style of manuscript have already updated meets PLOS ONE's style requirements. 

Question 2: Please confirm that you have included all items recommended in the PRISMA checklist including:

- the full electronic search strategy used to identify studies with all search terms and limits for at least one database.

- a Supplemental file of the results of the individual components of the quality assessment, not just the overall score, for each study included.

Response: Thanks for this suggestion, and the full electronic search strategy in PubMed have already listed in Methods section. Moreover, the individual components of the quality assessment for each study have already added in Supplementary file. 

Question 3: In your Data Availability statement, you have not specified where the minimal data set underlying the results described in your manuscript can be found. PLOS defines a study's minimal data set as the underlying data used to reach the conclusions drawn in the manuscript and any additional data required to replicate the reported study findings in their entirety. All PLOS journals require that the minimal data set be made fully available. For more information about our data policy, please see http://journals.plos.org/plosone/s/data-availability.

Response: Thanks for this suggestion, and the Data Availability statement have already changed into: “Data availability: All relevant data are within the paper and its Supporting information files.”

Question 4: PLOS requires an ORCID iD for the corresponding author in Editorial Manager on papers submitted after December 6th, 2016. Please ensure that you have an ORCID iD and that it is validated in Editorial Manager. To do this, go to ‘Update my Information’ (in the upper left-hand corner of the main menu), and click on the Fetch/Validate link next to the ORCID field. This will take you to the ORCID site and allow you to create a new iD or authenticate a pre-existing iD in Editorial Manager. Please see the following video for instructions on linking an ORCID iD to your Editorial Manager account: https://www.youtube.com/watch?v=_xcclfuvtxQ.

Response: Thanks for this suggestion, and the ORCID ID for the corresponding author will submit in Editorial Manager when upload the revised manuscript. 

Question 5: Please include captions for your Supporting Information files at the end of your manuscript, and update any in-text citations to match accordingly. Please see our Supporting Information guidelines for more information: http://journals.plos.org/plosone/s/supporting-information.

Response: Thanks for this suggestion, and the captions for your Supporting Information files have already added at the end of manuscript.

Reviewer #1: 

General comments: The authors report on a meta-analysis of an important topic - risk factors for dry eye syndrome. I thank the authors for their persistence with this large and complex analysis, however their analysis is limited by multiple methodologic and reporting concerns. The quality of evidence used to make these conclusions is also suspect, given the large proportion of cross-sectional studies that are susceptible to recall bias and variability in the adjustment for confounding factors across studies.

Response: As behalf of all co-authors, I would like to appreciate this referee due to thoughtful comments proposed by the peer review. After we revised the manuscript, those significant issues could be changed. Moreover, this comment have already addressed in Limitation section. 

Question 1: Methods, Abstract: The model used for meta-analysis should be specified, as well as the inclusion and exclusion criteria for study selection.

Response: Thanks for this suggestion, and the methods in abstract have already changed into: “PubMed, Embase, and the Cochrane library were systematically searched for studies investigated the risk factors for dry eye syndrome from their inception until September 2021. The odds ratio (OR) with 95% confidence interval (CI) was calculated using the random-effects model.”

Question 2: Conclusion, Abstract: The authors should elaborate on their conclusion section to discuss the implications of their findings and/or areas for future research.

Response: Thanks for this suggestion, and the conclusion in abstract have already changed into: “This study reported the comprehensive risk factors for dry eye syndrome, including demographic information, clinical characteristics, and comorbidities.”

Question 3: Introduction: expand on the risk factor ‘visual display terminal’ for readers who may be unfamiliar with this term.

Response: Thanks for this suggestion, and the following sentence have already added in the revised manuscript: “Moreover, the occupational risk factor of visual display terminal (VDT) use was related to the progression of DES, which could explained by decreases blink rate and increases the proportion of incomplete blinks could causing the increased exposure of the ocular surface to the environment.”

Question 4: Introduction: “However, the presence of other comorbidities and individuals’ characteristics on the risk of DES were not illustrated.” The authors should specify that there is conflicting or negative evidence for these other characteristics as opposed to these being uninvestigated characteristics.

Response: Thanks for this suggestion, and this sentence have already changed into: “However, whether the comorbidities of individuals could affect the risk of DES remained controversial. ”

Question 5: Methods: “The following search terms were used: ("Dry Eye Syndromes"[Mesh]) AND "Risk Factors"[Mesh].” The search strategy provided by the authors is quite simplistic and does not account for other formulations of the same concept found in these major databases. No text-based searches are also integrated to ensure maximal literature coverage.

Response: Thanks for this suggestion, and this sentence have already changed into: “PubMed, Embase, and the Cochrane library were systematically searched for eligible studies from their inception until September 2021, and using the following text word or Medical Subject Heading terms: "dry eye syndrome", "dry eye disease", "Keratoconjunctivitis Sicca", "Xerophthalmia", and "Risk Factors".”

Question 6: Methods: Did the authors investigate the gray literature as part of this systematic review?

Response: Thanks for this suggestion, and the following sentences have already added in Limitation section to address this comments: “Fourth, the analysis based on published articles, the gray literature and unpublished data were not available, and the publication bias was inevitable. Finally, the analysis using the pooled data, and the detailed analyses were restricted. ”

Question 7: Methods: specify with initials the authors that independently screened studies and extracted data.

Response: Thanks for this suggestion, and the initials of authors have already added in the revised manuscript for screened studies and extracted data.

Question 8: Methods: the authors note that all studies reporting on dry eye risk factors were considered. Were interventional studies assessing the impact of dry eye treatment also included?

Response: Thanks for this suggestion. The identified risk factors including demographic information, clinical characteristics, and comorbidities, while potential treatments in interventional studies were not included. We have already added the following sentence in Methods section: “Interventional study, animal study, review, and letter to editor was excluded”. 

Question 9: Methods: was the systematic review prospectively registered in the PROSPERO database?

Response: Thanks for this suggestion, this study was not registered in the PROSPERO, which have already addressed in Limitation section. 

Question 10: Methods: The authors note that studies which assessed risk factors in at least 3 cohorts were included. Some studies assessing risk factors for dry eye may have only focused on one risk factor and would therefore be excluded under the current methodology. This overly restrictive inclusion criterion seems arbitrary and unwarranted.

Response: Thanks for this suggestion. The main findings of this meta-analysis focused on quantitative analysis, and the conclusions obtained from 1 or 2 studies caused the variable of pooled results. Moreover, the screening criteria was referring to the following articles: Song P, Rudan D, Zhu Y, Fowkes FJI, Rahimi K, Fowkes FGR, Rudan I. Global, regional, and national prevalence and risk factors for peripheral artery disease in 2015: an updated systematic review and analysis. Lancet Glob Health. 2019 Aug;7(8):e1020-e1030. 

Question 11: Methods: Was a GRADE evaluation performed to assess the certainty of findings?

Response: Thanks for this suggestion, and this study contained cross-sectional, retrospective, and prospective observational studies, and the GRADE evaluation was not applied for certainty of findings. 

Question 12: Methods: “Identified risk factors for DES were analyzed based on the effect estimate, with its 95% CI, in 99 individual studies.” What effect estimate was considered? Elaborate on the exact methods used. For continuous variables like age, how were they categorized into binary groups for analysis?

Response: Thanks for this suggestion, and this sentence have already changed into: “Identified risk factors for DES were analyzed based on the OR, RR, or HR, with its 95% CI, in individual studies. Then the pooled ORs with 95%CI were calculated using the random-effects model.[16,17]”. Moreover, the categorized for continuous variables were referring to the original articles, which have already addressed in Limitation section. 

Question 13: Methods: Only one subgroup analysis based on country was considered in the analysis. There are likely multiple potential confounders in this analysis that can be investigated for using subgroup analysis, including severity of dry eye, individuals with dry eye secondary to another condition eg Sjogren’s, evaporative vs aqueous deficient dry eye, study design, etc.

Response: Thanks for this suggestion. In the planning stage, subgroup analysis should be performed according to study design, the severity of dry eye, and the type of dry eye. However, mostly included studies designed as cross-sectional studies. Moreover, DES was considered as investigated outcome, and the stratified data for the severity of dry eye, and the type of dry eye were not available in mostly included studies. 

Question 14: Methods, statistical analysis: specify the measure of association used to report the results - ie odds ratio with 95% confidence interval.

Response: Thanks for this suggestion, and this sentence have already changed into: “Identified risk factors for DES were analyzed based on the OR, RR, or HR, with its 95% CI, in individual studies. Then the pooled ORs with 95%CI were calculated using the random-effects model.[16,17]”

Question 15: Results: what proportion of results of studies was derived from a multivariable analysis that controlled for confounding parameters versus a univariable analysis?

Response: Thanks for this suggestion, and the adjusted factors have already abstracted and listed in Table 1. 

Question 16: Results: provide more information in the study characteristics section on the type and severity of dry eye, observational vs interventional studies, the recency of publication, the pooled gender and age distribution, and other relevant baseline parameters for this analysis.

Response: Thanks for this suggestion, the type and severity of DES were available in smaller number of included studies. Moreover, the baseline characteristics of included studies recruited individuals have already listed in Table 1.Finally, several sentences have already added in the second paragraph of Results section. 

Question 17: Results: expand on the results of the risk of bias assessment - what was the most frequent reason for downgrading the risk of bias?

Response: Thanks for this suggestion, and the following sentences have already added in the revised manuscript: “The quality of included studies mainly affect by the representativeness of the exposed cohort, and comparability on the basis of the design or analysis. ”

Question 18: Results: It would be important to clarify the definition of DES used in the analysis, and whether this definition was different across included studies.

Response: Thanks for this suggestion, and the definition of DES was based on included studies, and the definition was different among included studies. We have already address this comment in Limitation section. 

Question 19: Results: When reporting the results for each risk factor, it would be helpful to provide the proportion of the characteristic in the DES vs no DES groups, e.g. mean age, % female, race distribution, etc.

Response: Thanks for this suggestion. The analysis of this study based on pooled data, and the detailed analyses were restricted, which have already addressed in Limitation section. 

Question 20: Results: For endpoints that had significant statistical heterogeneity, was the heterogeneity changed in the subgroup analysis?

Response: Thanks for this suggestion, and the heterogeneity was not fully explained in the sensitivity and subgroup analyses, which have already addressed in Limitation section.

Question 21: Results: For glaucoma as a risk factor, it would important to clarify the therapy received, as this likely impacts the risk of DES.

Response: Thanks for this suggestion. The inclusion criteria was restricted risk factors were reported for ≥ 3 cohorts, and whether the therapies were adjusted have already listed in Table 1. 

Question 22: Results: For eye surgery as a risk factor, clarify what type of surgeries were considered.

Response: Thanks for this suggestion. The type of eye surgery was defined based on individual study, and the stratified data based on the type of eye surgery was not available. We have already addressed this comment in Limitation section. 

Question 23: Results: In the reporting of the various risk factors, certain risk factors are grouped with others in the same paragraph. It is unclear how the various risk factors were grouped. It would be helpful to add subheadings to clarify this.

Response: Thanks for this suggestion, and the subheadings have already added in the revised manuscript.

Question 24: Results: “systemic disease” as a risk factor is not helpful or clinically relevant. This needs to be specified as to which specific systemic disease was considered or otherwise removed from the analysis.

Response: Thanks for this suggestion. The systemic disease was defined based on individual study, and the stratified data based on systemic disease was not available. We have already addressed this comment in Limitation section. 

Question 25: Discussion: “we noted that other races versus white race were associated with an increased risk of DES, which is significantly related to the climate type.” Is race a risk factor independent of climate type? Elaborate on how climate type mediated this effect.

Response: Thanks for this suggestion. The climate type was not addressed in nearly all of included studies, and this comment have already addressed in Limitation section. 

Question 26: Discussion: “anti-hypertensive medication could increase the risk of DES”. What is the mechanism of this association?

Response: Thanks for this suggestion, and this sentence have already changed into: “A potential reason for this could be hypertension was not direct affect the risk of DES, while the use of anti-hypertensive medication could increase the risk of DES.[32] In addition, the risk of DES were not increased in hypertensive patients treated with anti-hypertensive medications, such as Angiotension Converting Enzyme inhibitors might play a protective role on the risk of DES.[33]”

Question 27: Discussion: “The role of rosacea could be explained by its significant relation to corneal neovascularization and perforation, which could induce vision loss and ocular comorbidities.[85]” This relation is poorly described. Rosacea is a well known risk factor for DES due to is pro-inflammatory effects that induce meibomian gland dysfunction and evaporative DES.

Response: Thanks for this suggestion, and this sentence have already changed into: “Rosacea is a well known risk factor for DES due to is pro-inflammatory effects that induce meibomian gland dysfunction and evaporative DES.[85]”

Question 28: Discussion: “Studies have already found that thyroid disease is significantly related to ocular surface damage, elevated lip aperture” - ‘elevated lip aperture’ should read ‘eyelid retraction’

Response: Thanks for this suggestion, and ‘elevated lip aperture’ have already changed into ‘eyelid retraction’ in the revised manuscript. 

Question 29: Discussion: a major limitation is that the impact of confounding variables is unaddressed by certain studies, and multivariable associations were only reported for certain risk factors. Given the large proportion of cross-sectional studies, recall bias is likely a significant issue in these results.

Response: Thanks for this suggestion. The adjusted factors have already mentioned in Table 1. Moreover, the following sentence have already added in Limitation section: “First, this study contained cross-sectional, retrospective, and prospective observational studies, and the results could affect by recall bias, which restricting the assessment of causality relationships between risk factors and DES.”

Question 30: Figure 2: p-value column - unsure why the p-values displayed in this way, as if multiple numbers are written on top of one another? What is the significance of the blue/gray background behind the forest plot for each endpoint?

Response: Thanks for this suggestion, and the Figure 2 have already changed in the revised manuscript. Moreover, the blue/gray background behind in the forest have already revised in Supplementary file.

Reviewer #2: 

General comments: This is a very well written meta-analysis regarding the potential risk factors for dry eye syndrome. Nevertheless, risk factors for DES are widely known, therefore, my question to the authors is:

In which way this meta analysis contributes to the literature already published regarding the risk factors for DES.

Response: We appreciate the reviewer given this kindly comments. Systematic reviews and meta-analyses are the most powerful tools in evaluating inconsistencies in risk factors. Our study designed as meta-analysis, and the inconsistent results for the risk factors of DES could determined. Moreover, the pooled effect estimates for the risk factors of DES could obtained based on large number of studies, and the results were stability.

---

## [Decision Letter · Decision Letter 1]

4 Apr 2022

PONE-D-21-35192R1Identified risk factors for dry eye syndrome: A systematic review and meta-analysisPLOS ONE

Dear Dr. Wei,

Thank you for submitting your manuscript to PLOS ONE. After careful consideration, we feel that it has merit but does not fully meet PLOS ONE’s publication criteria as it currently stands. Therefore, we invite you to submit a revised version of the manuscript that addresses the points raised during the review process.

We look forward to receiving your revised manuscript.

Kind regards,

Michael Mimouni

Academic Editor

PLOS ONE

Reviewers' comments:

Reviewer's Responses to Questions

**Comments to the Author**

1. If the authors have adequately addressed your comments raised in a previous round of review and you feel that this manuscript is now acceptable for publication, you may indicate that here to bypass the “Comments to the Author” section, enter your conflict of interest statement in the “Confidential to Editor” section, and submit your "Accept" recommendation.

Reviewer #1: (No Response)

Reviewer #2: All comments have been addressed

2. Is the manuscript technically sound, and do the data support the conclusions?

Reviewer #1: Yes

Reviewer #2: Yes

3. Has the statistical analysis been performed appropriately and rigorously? 

Reviewer #1: Yes

Reviewer #2: Yes

4. Have the authors made all data underlying the findings in their manuscript fully available?

Reviewer #1: Yes

Reviewer #2: Yes

5. Is the manuscript presented in an intelligible fashion and written in standard English?

Reviewer #1: Yes

Reviewer #2: Yes

6. Review Comments to the Author

Reviewer #1: The reviewer thanks the authors for their attempt at revising the manuscript. While improved, the manuscript still has a few limitations as noted below:

1. Conclusion, Abstract: “This study reported the comprehensive risk factors for dry eye syndrome, including demographic information, clinical characteristics, and comorbidities.” This sentence is not suitable as the last sentence of the abstract. Instead, the authors should discuss the implications of their research (i.e. why identification of these risk factors is important to improve clinical care) or alternatively discuss other areas of research that could be explored.

2. The manuscript should be thoroughly proofread by a native English speaker. For instance, this sentence contains multiple grammatical errors: “Moreover, the occupational risk factor of visual display terminal (VDT) use was related to the progression of DES, which could explained by decreases blink rate and increases the proportion of incomplete blinks could causing the increased exposure of the ocular surface to the environment”. A more appropriate sentence would be: “Moreover, the occupational risk factor of visual display terminal (VDT) use was related to the progression of DES, which could be explained by a decreased blink rate and increased proportion of incomplete blinks that could be caused by the increased exposure of the ocular surface to the environment”. In some sections, the language is so poor that it is uninterpretable: “which restricting the assessment of causality relationships between risk factors and DES”.

3. The search strategy is still insufficient to permit replication of the search by an independent researcher. The authors should specify the search strategy in a line-by-line format in a table that outlines clearly what term was searched, how terms were combined, and whether any restrictions were applied. The search should be updated to March 2022. In the table, the authors should highlight whether each term was a MeSH subject heading or a text based term.

4. The authors note that studies which assessed risk factors in at least 3 cohorts were included. The authors should cite the meta-analysis in Lancet Global Health that they followed for this methodology. I would advise changing ‘cohorts’ to ‘studies’ to make this sentence clearer in the Methods.

5. The authors note that “this study contained cross-sectional, retrospective, and prospective observational studies, and the GRADE evaluation was not applied for certainty of findings.” This reviewer believes it is important to conduct a GRADE evaluation for this meta-analysis especially because the evidence comes primarily from observational studies which are by their nature susceptible to bias. To provide readers with an indication of the certainty of evidence, the GRADE evaluation should be integrated.

6. The authors have discussed the variable definitions of DED in the limitations section, however in the baseline characteristics section of the results the authors should specify how different studies defined DED. This is important for readers to understand.

7. The sentence “we noted that other races versus white race were associated with an increased risk of DES, which is significantly related to the climate type” should be deleted because there is not evidence to support the notion that DES in different races is attributable to climate type as opposed to other factors.

Reviewer #2: Thank you for addressing my question and improving the shortcoming section. I do not have any more questions or comments to the authors.

7. PLOS authors have the option to publish the peer review history of their article (what does this mean?). If published, this will include your full peer review and any attached files.

Reviewer #1: No

Reviewer #2: No

---

## [Author Response · Author response to Decision Letter 1]

19 May 2022

Reviewer #1: 

General comments: The reviewer thanks the authors for their attempt at revising the manuscript. While improved, the manuscript still has a few limitations as noted below:

Response: As behalf of all co-authors, I would like to appreciate this referee due to thoughtful comments proposed by the peer review. After we revised the manuscript, those significant issues could be changed. 

Question 1: Conclusion, Abstract: “This study reported the comprehensive risk factors for dry eye syndrome, including demographic information, clinical characteristics, and comorbidities.” This sentence is not suitable as the last sentence of the abstract. Instead, the authors should discuss the implications of their research (i.e. why identification of these risk factors is important to improve clinical care) or alternatively discuss other areas of research that could be explored.

Response: Thanks for this suggestion, and the conclusion have already changed into: “This study reported risk factors for dry eye syndrome, and identified patients at high risk for dry eye syndrome.”

Question 2: The manuscript should be thoroughly proofread by a native English speaker. For instance, this sentence contains multiple grammatical errors: “Moreover, the occupational risk factor of visual display terminal (VDT) use was related to the progression of DES, which could explained by decreases blink rate and increases the proportion of incomplete blinks could causing the increased exposure of the ocular surface to the environment”. A more appropriate sentence would be: “Moreover, the occupational risk factor of visual display terminal (VDT) use was related to the progression of DES, which could be explained by a decreased blink rate and increased proportion of incomplete blinks that could be caused by the increased exposure of the ocular surface to the environment”. In some sections, the language is so poor that it is uninterpretable: “which restricting the assessment of causality relationships between risk factors and DES”.

Response: Thanks for this suggestion, and the English revision have already performed by Editage Company. 

Question 3: The search strategy is still insufficient to permit replication of the search by an independent researcher. The authors should specify the search strategy in a line-by-line format in a table that outlines clearly what term was searched, how terms were combined, and whether any restrictions were applied. The search should be updated to March 2022. In the table, the authors should highlight whether each term was a MeSH subject heading or a text based term.

Response: Thanks for this suggestion, and the details search strategy in PubMed have already added in S1 file. 

Question 4: The authors note that studies which assessed risk factors in at least 3 cohorts were included. The authors should cite the meta-analysis in Lancet Global Health that they followed for this methodology. I would advise changing ‘cohorts’ to ‘studies’ to make this sentence clearer in the Methods.

Response: Thanks for this suggestion, and this articles have already cited in the revised manuscript. Moreover, the ‘cohorts’ have already changed into ‘studies’.

Question 5: The authors note that “this study contained cross-sectional, retrospective, and prospective observational studies, and the GRADE evaluation was not applied for certainty of findings.” This reviewer believes it is important to conduct a GRADE evaluation for this meta-analysis especially because the evidence comes primarily from observational studies which are by their nature susceptible to bias. To provide readers with an indication of the certainty of evidence, the GRADE evaluation should be integrated.

Response: We acknowledge the importance of GRADE, while 39 of included studies designed as cross-sectional studies, and the causality relationships between risk factors and DES were not available. Moreover, the quality of included studies were assessed using the Newcastle-Ottawa Scale, and the results are listed in S1 table.

Question 6: The authors have discussed the variable definitions of DED in the limitations section, however in the baseline characteristics section of the results the authors should specify how different studies defined DED. This is important for readers to understand.

Response: Thanks for this suggestion, and the following sentence have already added in Results section: “The DES definition based on questionnaire were reported in 33 studies, 10 studies used TBUT, ST, or FSS defined DES, 3 studies applied ICD9 code and the remaining 2 studies used clinician-diagnosed defined DES.”

Question 7: The sentence “we noted that other races versus white race were associated with an increased risk of DES, which is significantly related to the climate type” should be deleted because there is not evidence to support the notion that DES in different races is attributable to climate type as opposed to other factors.

Response: Thanks for this suggestion, and this sentence have already removed in the revised manuscript.

Reviewer #2: 

General comments: Thank you for addressing my question and improving the shortcoming section. I do not have any more questions or comments to the authors. 

Response: We appreciate the reviewer given this kindly comments.

---

## [Decision Letter · Decision Letter 2]

28 Jun 2022

Identified risk factors for dry eye syndrome: A systematic review and meta-analysis

PONE-D-21-35192R2

Dear Dr. Wei,

We’re pleased to inform you that your manuscript has been judged scientifically suitable for publication and will be formally accepted for publication once it meets all outstanding technical requirements.

Kind regards,

Michael Mimouni

Academic Editor

PLOS ONE

Additional Editor Comments (optional):

Reviewers' comments:

Reviewer's Responses to Questions

**Comments to the Author**

1. If the authors have adequately addressed your comments raised in a previous round of review and you feel that this manuscript is now acceptable for publication, you may indicate that here to bypass the “Comments to the Author” section, enter your conflict of interest statement in the “Confidential to Editor” section, and submit your "Accept" recommendation.

Reviewer #1: All comments have been addressed

Reviewer #2: All comments have been addressed

2. Is the manuscript technically sound, and do the data support the conclusions?

Reviewer #1: Yes

Reviewer #2: Yes

3. Has the statistical analysis been performed appropriately and rigorously? 

Reviewer #1: Yes

Reviewer #2: Yes

4. Have the authors made all data underlying the findings in their manuscript fully available?

Reviewer #1: No

Reviewer #2: Yes

5. Is the manuscript presented in an intelligible fashion and written in standard English?

Reviewer #1: Yes

Reviewer #2: Yes

6. Review Comments to the Author

Reviewer #1: The reviewer thanks the author for revising the manuscript based on earlier feedback. I have no further comments.

Reviewer #2: I do not have more comments for the authors. Thank you for answering all the reviewers questions and improving the script.

7. PLOS authors have the option to publish the peer review history of their article (what does this mean?). If published, this will include your full peer review and any attached files.

Reviewer #1: No

Reviewer #2: No

---

## [Editor Report · Acceptance letter]

10 Aug 2022

PONE-D-21-35192R2 

Identified risk factors for dry eye syndrome: A systematic review and meta-analysis 

Dear Dr. Wei:

I'm pleased to inform you that your manuscript has been deemed suitable for publication in PLOS ONE. Congratulations! Your manuscript is now with our production department. 

Kind regards, 

on behalf of

Dr. Michael Mimouni 

Academic Editor

PLOS ONE